# Yeast encapsulation of photosensitive insecticides increases toxicity against mosquito larvae while protecting microorganisms

**Cole J. Meier, Veronica R. Wrobleski, Julián F. Hillyer**  *

Department of Biological Sciences, Vanderbilt University, Nashville, TN, United States of America

* julian.hillyer@vanderbilt.edu

## Abstract

An important defense against the deadly diseases that mosquitoes transmit is the application of insecticides that reduce mosquito populations. Unfortunately, the evolution and subsequent spread of insecticide resistance has decreased their efficacy. Therefore, new mosquito control strategies are needed. One class of larvicides, known as photosensitive insecticides, or PSIs, kills larvae via light-activated oxidative damage. PSIs are promising larvicides because of their high larvicidal efficacy, rapid photodegradation, inexpensive cost, and mechanism that is dissimilar to other insecticide classes. We explored a novel delivery strategy for increasing both the larvicidal efficiency and environmental biocompatibility of PSIs, known as yeast encapsulation. Using the PSIs, curcumin and methylene blue, we measured the survival of *Anopheles gambiae* larvae and *Escherichia coli* following exposure to either non-encapsulated or yeast-encapsulated PSIs and a photoperiod. Yeast encapsulation increased the phototoxicity of both curcumin and methylene blue against mosquito larvae, likely by increasing ingestion. Furthermore, yeast encapsulation protected *E. coli* from the phototoxicity of yeast-encapsulated curcumin, but not yeast-encapsulated methylene blue. Yeast encapsulation increases the larvicidal efficacy of a PSI while also increasing biocompatibility. Therefore, yeast encapsulation of PSIs is a promising insecticide delivery strategy for mosquito control.

## Introduction

Mosquitoes transmit deadly diseases, making them a necessary target for pest control [1, 2]. The most common form of mosquito control is the application of chemical insecticides that target the adult life stage [2]. However, the development and spread of insecticide resistance threatens their efficacy [1–3]. Therefore, new mosquito control strategies are desperately needed.

One promising option for mosquito control is the use of photosensitive insecticides, or PSIs. PSIs are inexpensive molecules that generate reactive oxygen species (ROS) when

**Funding:** This research was funded by Vanderbilt University (https://www.vanderbilt.edu) institutional funds to JFH. The funders had no role in study design, data collection and analysis, decision to publish, or preparation of the manuscript.

**Competing interests:** The authors have declared that no competing interests exist.

activated by light [4–6]. Because mosquito larvae are filter feeders, they readily consume PSIs that are applied to their aquatic environment [7, 8]. Once ingested by a larva, PSIs are photoactivated and the ROS that they produce indiscriminately damage any surrounding tissue, eventually resulting in larval death [5, 6, 9, 10]. PSIs offer several advantages relative to other mosquito control options: (i) by targeting the larval juvenile life stage of the mosquito, PSIs eliminate both the disease transmitting potential of mosquitoes and their reproductive potential [5]; (ii) PSIs rapidly photodegrade hours after their application, so deleterious environmental accumulation does not occur [5, 11]; (iii) the non-specificity of ROS-mediated damage means that mosquitoes are unable to develop target-site resistance against a PSI [5, 12, 13]; and (iv) PSIs are highly toxic to evolutionarily diverse larvae, including the major disease vectors *Aedes aegypti*, *Aedes albopictus*, *Culex quinquefasciatus*, and *Anopheles gambiae* [7, 8, 14–19].

Despite these advantages, hurdles remain regarding the environmental biocompatibility of PSIs. Although PSIs have no apparent toxicity against opaque organisms that internalize a PSI, as demonstrated by their therapeutic use in humans [20–22], translucent organisms likely remain susceptible. Size may buffer larger translucent organisms from PSI stress, as seen by curcumin not being toxic against adult *Danio rerio* (zebrafish) [17], but unicellular organisms are susceptible to PSI-mediated ROS damage [12, 13, 23–25]. This is concerning because aquatic microorganisms perform key functions in the ecosystem, such as nutrient processing, oxygen synthesis, organic matter breakdown, and biomass production [26, 27]. Their disturbance decreases water quality and the availability of some nutrients needed by other organisms [28]. Therefore, their disruption reduces biodiversity and environmental health.

One potential way to increase the environmental biocompatibility of PSIs is to encapsulate them in yeast. Yeast encapsulation is a process where prolonged agitation and high heat forces exogenous molecules into a yeast cell, a process that also kills the yeast [29]. Yeast encapsulation improves the solubility and stability of xenobiotics [30], and in the context of a larvicide, an additional benefit is that the larvicide is sequestered from the environment because it is contained within the yeast cell. Because mosquito larvae readily digest yeast cells [31], yeast-encapsulated orange oil is toxic against *Aedes aegypti* larvae [32–34]. Yeast encapsulation of PSIs may also protect microorganisms because the yeast cell may sequester the ROS produced during photoactivation in the aquatic environment. Although other translucent insects that feed on yeast could remain susceptible to yeast-encapsulated PSIs, yeast encapsulation is a potential strategy to decrease the phototoxicity of PSIs against non-target species.

Because larvae actively feed on yeast, we hypothesize that yeast-encapsulated PSIs are more toxic to larvae than non-encapsulated PSIs. This hypothesis is supported by the finding that adding PSIs to a larval environment that includes food can render them more toxic to mosquitoes, presumably because larval feeding is stimulated [8]. Here, we examined whether yeast encapsulation affects the toxicity of two PSIs, curcumin and methylene blue, against larvae of the mosquito, *An. gambiae*, and the bacterium, *Escherichia coli*. We discovered that yeast encapsulation increases the toxicity of these PSIs against mosquito larvae and decreases the toxicity of curcumin, but not methylene blue, against *E. coli*. We conclude that yeast encapsulation increases the larvicidal efficacy and environmental biocompatibility of PSIs.

## Materials and methods

### Larval rearing and maintenance

*Anopheles gambiae* Giles 1902 *sensu stricto* (G3 strain; Diptera: Culicidae) were raised and maintained as described [35]. Mosquitoes were hatched and reared in an environmental chamber at 27˚C and 75% relative humidity, under a 12 hr:12 hr ambient light:dark cycle. Larvae were fed koi food and baker's yeast, and all experiments were conducted inside the

environmental chamber and initiated with 4th instar larvae. Fourth instar larvae were selected for experimentation because of their larger size, and so that the findings could be correlated to our previously published research.

## Preparation of yeast-encapsulated photosensitive insecticides

A stock solution of 2.5 mM (0.8 mg/mL) trihydrate methylene blue (Sigma-Aldrich, St Louis, MO, USA) was prepared in deionized water. Because of curcumin's poor solubility in water [36, 37], a stock solution of 2.9 mM (1.1 mg/mL) of curcumin (TCI America, Portland, OR, USA) was prepared in ethanol (Fisher Scientific, Hampton, NH, USA). These stock solutions were wrapped in aluminum foil and stored in the dark, at room temperature.

Yeast encapsulation of PSIs was conducted by adapting the methods of Paramera et al [38]. Briefly, 1 g of Saf-Instant Red dry yeast (Lesaffre, Marcq-en-Barœul, France), 500 mL of deionized water, and either 20 μM methylene blue or 100 μM curcumin were mixed in a 1000 mL Erlenmeyer flask. The solutions were placed in a shaking incubator and incubated overnight at 230 RPM and 40°C. Then, 45 mL aliquots were transferred to 50 mL tubes and centrifuged at 2,000 RCF for 5 min at 4°C. The supernatant containing the non-encapsulated PSI was removed without disturbing the pelleted yeast, and 45 mL of ice-cold deionized water was added to the tube and vortexed briefly. The above centrifugation and washing steps were repeated 2–3 times to remove any non-encapsulated PSI, at which point the supernatant was visibly clear (Fig 1). The washed yeast-encapsulated PSI was resuspended in 10 mL of ice-cold deionized water, wrapped in aluminum foil, and stored at 4°C until use.

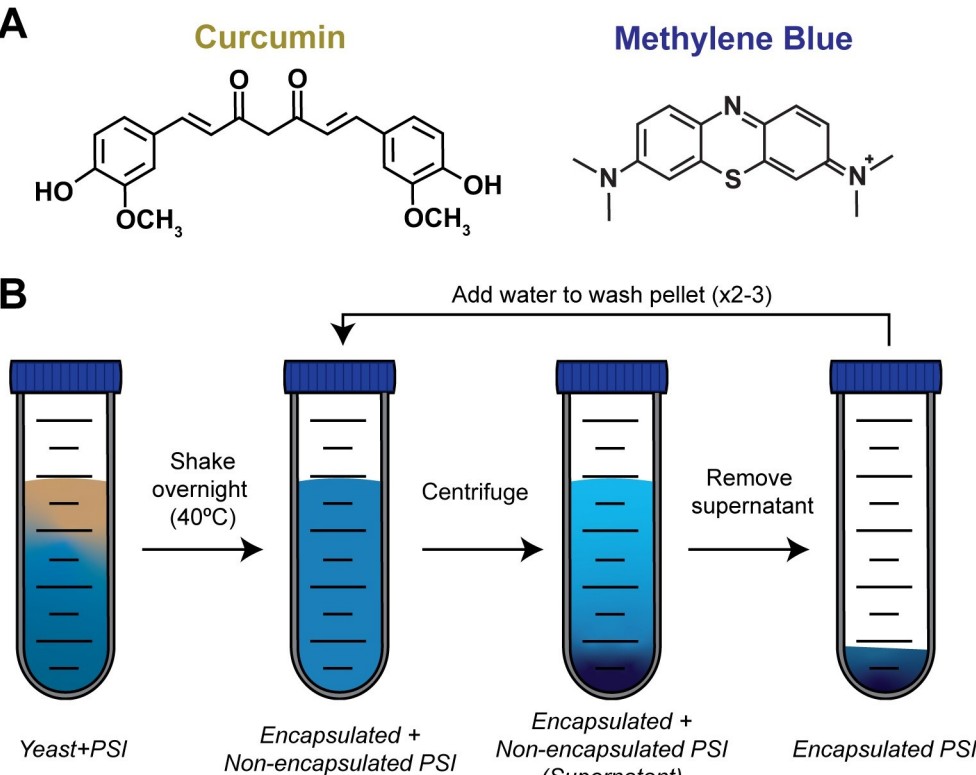

**Fig 1. Yeast-encapsulation of a PSI.** (A) Molecular structure of curcumin and methylene blue. (B) Protocol for yeast encapsulation and washing steps of a PSI.

### Larval incubation, photoactivation, survival, and microscopic examination

PSI incubations to determine larvicidal activity were conducted as we have previously described [8, 35]. Fourth instar larvae were transferred from a mass larval rearing container to a clear 6-well plate containing room temperature deionized water. Larvae were then transferred to clean water two more times to remove particulates that had been transferred from the mass rearing container. Once rinsed, 10 larvae were added to each well of a 6-well plate, excess water was removed, and 5 mL of water was added to each well in addition to either ethanol, non-encapsulated methylene blue or curcumin, or yeast-encapsulated methylene blue or curcumin.

Well plates were covered in aluminum foil and incubated at 27˚C for 2 hr. Larvae were incubated between Zeitgeber time 0 and 6 (where Zeitgeber time 0 indicates the beginning of the light phase in a 12 hr:12 hr light dark cycle) because pupation is minimal during this time, and we discovered that pupation protects larvae from PSI toxicity [35]. Following this incubation, larval survival was measured via a mechanical test that used a plastic pipette as a stimulus probe; larvae that responded to the probe were alive whereas those that did not were dead. Well plates were then either kept in aluminum foil to measure survival in the dark or transferred to a white surface that was ~15 cm beneath a 5000 Lumen LED lamp (5000 Lumen LED Work Light, Husky Corporation, Pacific, MO) to measure survival in the light (lamp illuminates ~1,250 lux). For plates placed underneath the lamp, the lids were removed, the lamp turned on to initiate the photoperiod, and survival was measured every 20 min for 2 hr. Larval survival was measured using the mechanical test, with the well plates temporarily removed from the photoperiod for no more than 2 min every 20 min. For larvae kept in the dark, survival was only measured upon the conclusion of the initial incubation, and at 2 hr following the second darkness incubation (which coincided with the conclusion of the photoperiod in the light-activated plates). All well plates and their lids were then transferred to the environmental chamber under ambient lighting conditions for 22 hr, and survival was measured one final time. The ambient lighting of the environmental chamber is insufficient to activate the PSIs but maintains the mosquitoes' diurnal cycle.

To determine larval sensitivity to ethanol, larvae were added to 5 mL of 0.002%, 0.004%, 0.01%, 0.02%, or 0.04% of ethanol diluted in water. To determine the baseline survival of larvae exposed to curcumin, larvae were added to 3 µM, 6 µM, 9 µM, and 12 µM curcumin in 5 mL of water. This baseline experiment was not conducted for methylene blue because we already established baseline survival for this PSI [8, 35]. To determine the effects of yeast encapsulation on PSI larvicidal activity, larvae were added to either 3 µM and 9 µM of non-encapsulated and yeast-encapsulated curcumin, or 0.5 µM and 1 µM of non-encapsulated or yeast-encapsulated methylene blue. To determine whether a toxic amount of unencapsulated PSI remained in our yeast encapsulate solutions, larvae were added to 5 mL of water that contained 50 µL, 200 µL, or 500 µL of the supernatant wash from the final washing step (described above; Fig 1) of yeast-encapsulated curcumin or methylene blue.

For microscopic examinations, larvae were viewed under oblique coherent contrast trans-illumination using a Nikon SMZ 1500 stereomicroscope (Nikon Corporation, Tokyo, Japan). Micrographs were acquired using a Nikon Digital Sight DS-Fi1 5 MP CCD Color Camera and Nikon's Advanced Research NIS Elements software.

### *E. coli* growth, photoactivation, and survival

Tetracycline resistant, GFP-expressing *E. coli* (modified DH5$\alpha$; GFP-*E. coli)* were grown overnight in Luria-Bertani (LB) media at 37˚C in a shaking incubator (New Brunswick Scientific,

Edison, NJ, USA). The optical density of the *E. coli* culture was then measured spectrophoto-metrically and diluted with LB to $OD_{600} = 2$.

To determine whether the LED light treatment is toxic to *E. coli*, the culture was further diluted 1:100 with LB, and 12.5 μL of the diluted culture was added to deionized room temperature water for a total volume of 5 mL in each well of a clear 6-well plate. The well plate was then transferred to the 27˚C environmental chamber and either wrapped in aluminum foil (darkness treatment) or placed underneath the 5000 Lumen lamp (light treatment) for either 30 min or 2 hr. Then, 1 mL was taken from each well, vortexed, and 40 μL of the solution was plated on an LB plate containing tetracycline and grown overnight at 37˚C. The number of colony forming units (CFUs) that grew on each plate was then counted as a measure of *E. coli* survival.

To determine whether non-encapsulated or yeast-encapsulated PSIs are toxic to *E. coli*, either 9 μM of non-encapsulated or yeast-encapsulated curcumin, or 1 μM of non-encapsulated or yeast-encapsulated methylene blue was added to the wells in the 6-well plate. Well plates were transferred to the 27˚C environmental chamber and either wrapped in aluminum foil (darkness treatment) or placed underneath the 5000 Lumen lamp for 30 min (light treatment). After 30 min, *E. coli* survival was measured using the plating method described above.

## Sampling and statistical analysis

For each parameter measured, each treatment was evaluated over a minimum of 3 independent trials. For larval survival experiments, at least 90 larvae were used per treatment, with the larvae originating from at least 3 different egg batches. The exact sample sizes for each parameter measured are presented in the figures. The few larvae that died during the initial dark incubation period, or pupated before the conclusion of the photoperiod, were omitted from analysis. This was done because death during the initial darkness incubation period is not due to PSI phototoxicity, and pupation protects from PSI toxicity [35]. All trials were conducted using duplicate well plates; one well plate was exposed to a photoperiod whereas the other plate was maintained in darkness. The one exception was trials investigating the toxicity of yeast-encapsulated PSI supernatant, where no well plate was maintained in darkness. For *E. coli* survival experiments, 3 independent trials were conducted, each with 2–3 independent samples per treatment.

Larval survival curves were compared using the Logrank Mantel Cox test. The *E. coli* CFU counts were compared using either the Mann-Whitney U-test when comparing two groups, or the Kruskal-Wallis test when comparing more than two groups. All data analysis was completed in GraphPad Prism version 9.4.1, and differences were deemed significant at $P < 0.05$. All data collected in this manuscript are included in (S1 Table).

## Results

### Curcumin is phototoxic to larvae

To explore the effects of yeast encapsulation on PSI toxicity, we first needed to know how phototoxic each PSI is to *An. gambiae* larvae when non-encapsulated. We recently demonstrated that methylene blue in the presence of a photoperiod is lethal to *An. gambiae* larvae at concentrations >2 μm, mildly toxic at 1 μm, and non-toxic at 0.5 μM [8, 35]. This makes 0.5 μM and 1 μM methylene blue ideal for measuring whether yeast encapsulation increases phototoxicity. The phototoxicity of curcumin, however, has only been investigated in *Aedes aegypti* larvae [7, 16, 17]. Therefore, we first aimed to determine the phototoxicity of non-encapsulated curcumin against *An. gambiae* larvae.

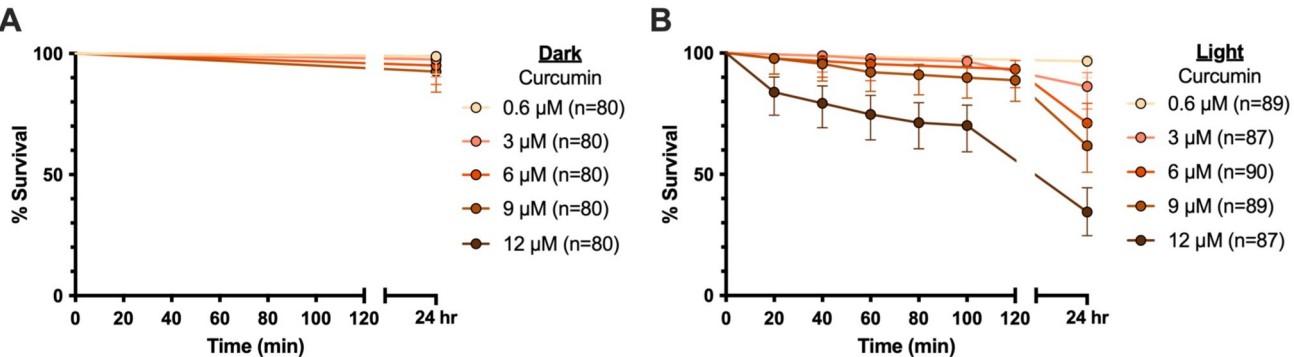

**Fig 2. Survival of mosquito larvae exposed to curcumin.** Larval survival was measured following incubation with either 0.6 μM, 3 μM, 6 μM, 9 μM, or 12 μM of curcumin in the dark for 2 hr, followed by 2 hr of either continued darkness (A) or photoactivation (B). All larvae were then exposed to 22 hr of ambient lighting (insufficient for photoactivation). Time zero corresponds to the initiation of the second period of continued darkness (A) or the 2 hr photoperiod (B). Whiskers indicate the 95% confidence interval, and n indicates number of mosquitoes.

Due to curcumin's poor solubility in water [36, 37], we prepared curcumin stock solutions in ethanol. Therefore, before measuring the phototoxicity of non-encapsulated curcumin, we set to determine the maximum amount of ethanol that larvae can tolerate. We found that all larvae survived by 24 hr when exposed to 0.004% ethanol; however, at 0.01% ethanol, 15% larvae died by 24 hours (S1 Fig). Therefore, using 0.004% as our maximum amount of ethanol added with curcumin, we measured the survival of larvae following exposure to 0.6 μM, 3 μM, 6 μM, 9 μM, and 12 μM curcumin in the dark (Fig 2A). All larvae survived, confirming that in the darkness curcumin is not toxic to larvae.

We next examined whether curcumin is toxic to larvae when activated via a photoperiod (Fig 2B). Exposure to 0.6 μM curcumin was not toxic to larvae, with 97% surviving by 24 hrs. However, ≥3 μM curcumin was toxic to larvae. Only 86% of larvae survived exposure to 3 μM curcumin and a photoperiod. Larval survival continued to decrease as the curcumin concentration increased, with 34% survival after exposure to 12 μM curcumin—the highest concentration tested. These findings demonstrate that curcumin's phototoxicity is lethal to larvae.

## Yeast encapsulation of curcumin and methylene blue increases the phototoxicity against larvae

To determine whether yeast encapsulation affects the toxicity of PSIs against larvae, we exposed larvae to either non-encapsulated or yeast-encapsulated curcumin or methylene blue. For both curcumin and methylene blue, we investigated concentrations that result in low (3 μM curcumin and 0.5 μM methylene blue) and medium (9 μM curcumin and 1 μM methylene blue) larval mortality when the non-encapsulated PSI is photoactivated. As a negative control, larvae were exposed to the same treatments in the absence of a photoperiod; in these larvae no meaningful mortality (<5%) was observed (S2 Fig). Additionally, to verify that our yeast-encapsulated PSI solutions did not contain a toxic amount of free, non-encapsulated PSI, we incubated larvae in the supernatant collected from our final wash of the yeast encapsulation protocol and exposed them to a photoperiod; no meaningful mortality was observed (S3 Fig).

Yeast-encapsulated curcumin was more toxic to larvae than non-encapsulated curcumin (Fig 3A and 3B). After exposure to 3 μM non-encapsulated curcumin and a photoperiod, 92% of larvae survived by 24 hr, but after exposure to an equivalent concentration of yeast-encapsulated curcumin, only 3% of larvae survived and the median survival time was reached at 20

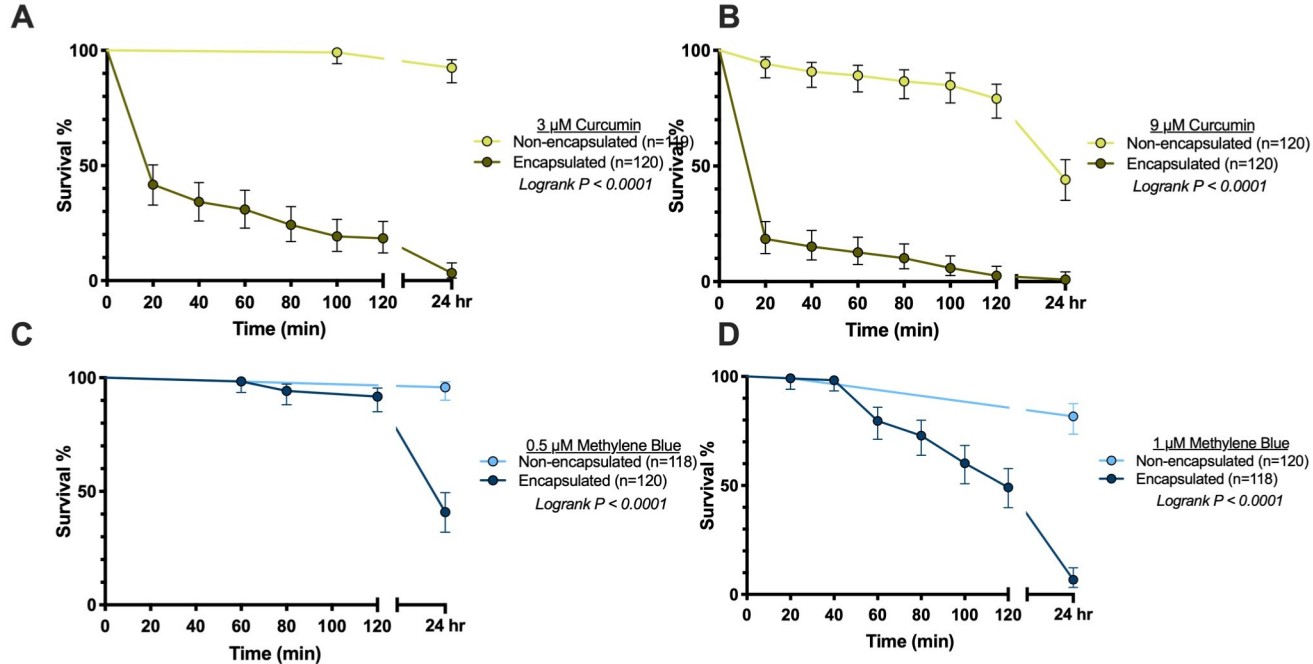

**Fig 3. Survival of mosquito larvae following exposure to non-encapsulated or yeast-encapsulated curcumin and methylene blue.** Larval survival was measured after incubation with either 3 µM curcumin (A), 9 µM curcumin (B), 0.5 µM methylene blue (C) or 1 µM methylene blue (D) that was either non-encapsulated or encapsulated in yeast. Larvae were exposed for 2 hr in continued darkness, followed by 2 hr of photoactivation and 22 hr of ambient lighting (insufficient for photoactivation). Time zero corresponds the initiation of the photoperiod. Whiskers indicate the 95% confidence interval, and n indicates the number of mosquitoes.

min after the initiation of the photoperiod. After exposure to 9 µM non-encapsulated curcumin and a photoperiod, 44% of larvae survived by 24 hr and the median survival time was reached sometime between 2 hr and 24 hr, but after exposure to an equivalent concentration of yeast-encapsulated curcumin and a photoperiod, only 1% of larvae survived and the median survival time was 20 min.

Similar to curcumin, yeast-encapsulated methylene blue was more toxic to larvae than non-encapsulated methylene blue (Fig 3C and 3D). After exposure to 0.5 µM non-encapsulated methylene blue and a photoperiod, 96% of larvae survived by 24 hr, but following exposure to an equivalent concentration of yeast-encapsulated methylene blue, only 41% of larvae survived and the median survival time was sometime between 2 hr and 24 hr. After exposure to 1 µM non-encapsulated methylene blue and a photoperiod, 82% of larvae survived by 24 hr, but following exposure to an equivalent concentration of yeast-encapsulated methylene blue and a photoperiod, only 7% of larvae survived and the median survival time was 120 min. Altogether, yeast encapsulation greatly increases the toxicity of curcumin and methylene blue against larvae.

## Yeast encapsulation increases larval ingestion of curcumin and methylene blue

Adding larval food to the water can increase the toxicity of PSIs, presumably by stimulating larval feeding [35]. Because yeast encapsulation enhances the phototoxicity of PSIs, we suspected that larvae consume more PSIs when they are encapsulated by yeast. To qualitatively determine this, we imaged larvae following incubation with the same amount of either non-encapsulated or yeast-encapsulated PSI, and observed the ingestion of curcumin and

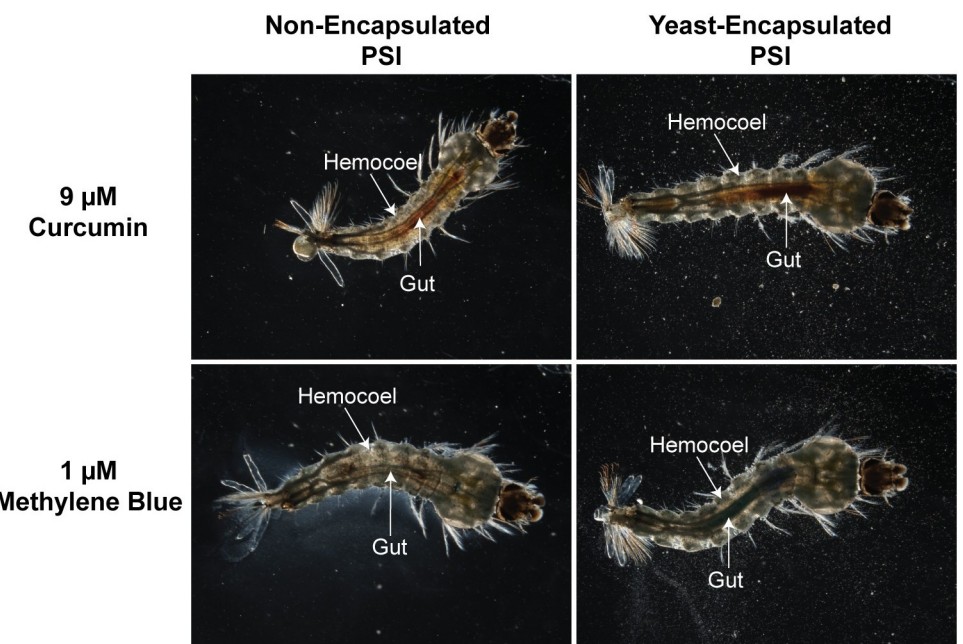

**Fig 4. Ingestion of non-encapsulated and yeast-encapsulated curcumin and methylene blue by mosquito larvae during a 2 hr incubation.**

methylene blue by visualizing the presence of yellow or blue dye within the larvae, respectively (Fig 4). Compared to the non-encapsulated PSIs, yeast encapsulation increases the amount of PSI that is ingested by larvae. Moreover, yeast-encapsulated PSIs appeared confined to the larval gut instead of distributing throughout the hemocoel. From this, we infer that yeast encapsulation increases the phototoxicity of PSIs by enhancing larval feeding.

## Yeast encapsulation decreases phototoxicity of curcumin against *E. coli*

Photosensitive molecules are phototoxic to microorganisms [25]. To determine how yeast encapsulation affects the toxicity of curcumin and methylene blue against microorganisms, we measured *E. coli* survival after exposure to non-encapsulated or yeast-encapsulated curcumin and methylene blue. First, we assessed whether a photoperiod itself, in the absence of a PSI, is toxic to *E. coli*. We found that a 2 hr photoperiod is toxic to *E. coli* (Mann-Whitney $P = 0.0005$) but that a 30 min photoperiod is not (Mann-Whitney $P = 0.5457$; S4 Fig). Second, we assessed whether non-encapsulated or yeast-encapsulated PSIs in the absence of a photoperiod are toxic to *E. coli*. We found that non-photoactivated PSIs do not affect *E. coli* survival (Kruskal-Wallis $P = 0.7502$; S5 Fig). Hence, we conducted follow-up experiments using a 30 min photoperiod because it does not significantly affect *E. coli*.

Yeast encapsulation protected *E. coli* from curcumin toxicity (Fig 5A). As predicted, 9 µM non-encapsulated curcumin was toxic to *E. coli* when photoactivated (Mann-Whitney $P < 0.0001$). However, when *E. coli* were exposed to an equivalent amount of yeast-encapsulated curcumin, the survival of *E. coli* was similar between the photoactivated and non-photoactivated treatments (Mann-Whitney $P = 0.9551$).

Yeast encapsulation did not protect *E. coli* from methylene blue toxicity (Fig 5B). As predicted, 1 µM non-encapsulated methylene blue was toxic to *E. coli* when photoactivated (Mann-Whitney $P < 0.0001$). Surprisingly, 1 µM of yeast-encapsulated methylene blue was just as toxic to *E. coli* as non-encapsulated methylene blue (Mann-Whitney $P < 0.0001$).

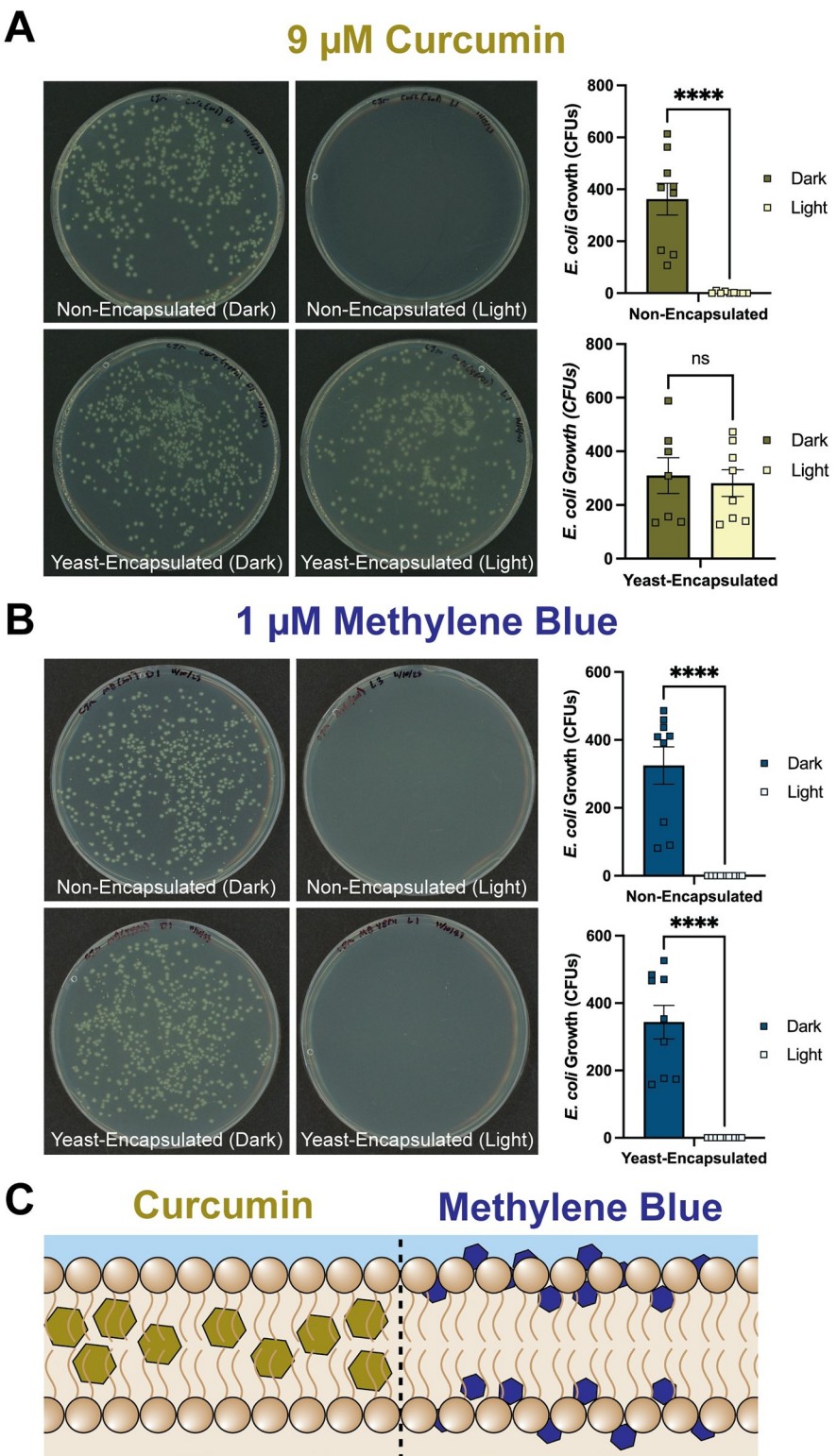

**Fig 5. Survival of *E. coli* following exposure to either non-encapsulated or yeast-encapsulated curcumin and methylene blue.** *E. coli* were exposed to 9 μM of curcumin (A) or 1 μM methylene blue (B) that was either non-encapsulated or encapsulated in yeast. *E. coli* were either kept in the dark or exposed to a 30 min photoperiod, and then plated, grown overnight, and colony forming units (CFUs) counted to determine bacterial growth and survival. Column heights mark the mean, whiskers denote the S.E.M, and squares are the individual samples. Data were

analyzed using the Mann-Whitney U-test (ns, $P > 0.05$; ****, $P < 0.0001$). (C) Model of PSI distribution within a yeast cell for curcumin and methylene blue.

Altogether this suggests that yeast encapsulation is a viable strategy to decrease the toxicity of some PSIs—like curcumin—against microorganisms, but that this protection is not universal for all PSIs.

## Discussion

PSIs are a promising class of larvicide that, when activated by light, kills larvae via oxidative damage [5]. Because the larvicidal activity is caused by indiscriminate oxidative damage, resistance against this type of insecticide is less likely to evolve. However, oxidative damage can also kill non-target species, and a particular concern with any insecticide is the health of the aquatic microbial community. Here, we encapsulated PSIs in yeast to determine whether this process alters larvicidal activity while protecting microorganisms. We discovered that (i) yeast encapsulation increases the toxicity of PSIs against larvae, likely because of increased ingestion of the larvicide, and (ii) yeast encapsulation protects microorganisms from curcumin, but methylene blue (Fig 6). Overall, yeast encapsulation is a promising strategy to increase the larvicidal efficacy and environmental biocompatibility of some PSIs to control mosquito populations.

For PSIs to kill a larva, they first need to be ingested. Once ingested, the ROS generated inside the larva indiscriminately damage larval tissues, and if sufficient damage ensues, the larva dies [5, 6, 9, 10]. Therefore, the larvicidal efficiency of PSIs is directly influenced by the amount of PSI that is ingested. When food is present with excess PSI, the toxicity of PSIs increases by stimulating feeding and ingestion of the PSI; however, without an excess of PSI, food can sequester the PSIs and decrease their toxicity [8]. Here, we found that the encapsulation of curcumin and methylene blue in yeast increases the larvicidal toxicity of these PSIs relative to non-encapsulated PSIs. This suggests that, despite yeast encapsulation having the potential to sequester the ROS generated by a PSI, a more dramatic effect on toxicity is the associated increase in larval feeding of yeast-encapsulated PSI. This is supported by the observation that more yeast-encapsulated PSI are ingested than nonencapsulated PSI. Therefore, yeast encapsulation is an efficient way to maximize larvicidal efficiency of PSIs.

When non-encapsulated curcumin and methylene blue are ingested by larvae, they permeate out of the gut and into the hemocoel [8, 17]. However, we observed that yeast- encapsulated curcumin and methylene blue largely remain confined to the gut. It is possible that the binding of PSIs to yeast components makes them too large to diffuse across the gut epithelium. Therefore, it is surprising that yeast encapsulation increases the toxicity of these PSIs because their high toxicity has been attributed in part to their ability to cross the gut epithelium and damage tissues throughout the larva [8]. We hypothesize that the markedly greater ingestion of yeast-encapsulated PSIs compared to non-encapsulated PSIs compensates for their spatially restricted damage. However, it is possible that yeast encapsulation may not be as toxic in environments already saturated with larval food. Under these conditions, larvae may (i) become satiated after feeding on food absent of PSI, (ii) fail to encounter the PSI due to diminished bioavailability amidst the organic particulates, or (iii) may not receive sufficient irradiation from sunlight to photoactivate the PSI. When the non-encapsulated PSI, rose bengal, was applied to cesspits with an abundance of organic particulates, for example, the average larval mortality fell from 97% when applied under laboratory conditions, to only 3% [15]; however, the exact reason for this decrease in efficacy is unclear, and may be due to light restriction. So,

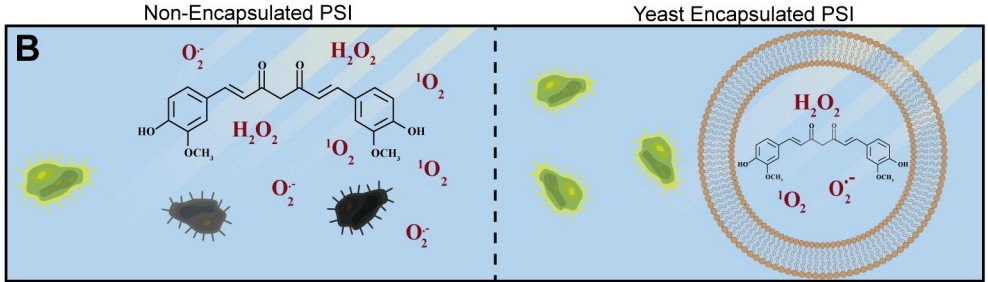

*Photosensitive insecticides (PSI) can be encapsulated in yeast.*

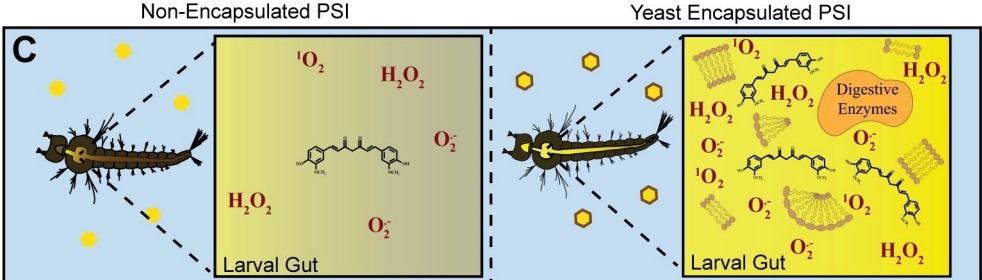

*Yeast encapsulation protects microorganisms from PSI phototoxicity.*

*Yeast encapsulated PSIs are more readily consumed by larvae;*
*the yeast encapsulate is then digested, and the PSI is released.*

**Fig 6. Yeast encapsulation of a PSI can decrease the phototoxicity against microorganisms and increase the phototoxicity against mosquito larvae.** (A) Yeast encapsulation of a PSI. (B) A non-encapsulated PSI is phototoxic to microorganisms via ROS, but when encapsulated, the PSI is sequestered away from environmental microorganisms. (C) Non-encapsulated PSIs are phototoxic to mosquito larvae, but the phototoxicity increases when the PSI is encapsulated in yeast. This is because larvae ingest more PSI when encapsulated in yeast than when non-encapsulated, and upon digestion of the yeast, the PSI is freed and damages surrounding tissue via ROS.

additional work is needed to examine the larvicidal efficacy of yeast-encapsulated PSIs in environments with varying amounts of detritus.

Yeast encapsulation increases the larvicidal activity of PSIs, and this finding is particularly striking because in this study we underestimate the toxicity of yeast-encapsulated PSIs. Encapsulation efficiency, or the amount of xenobiotic that enters the yeast cells relative to the amount that remains in solution, is affected by factors such as plasmolysis of the cells, temperature, solvent, and mass ratio of xenobiotic to yeast [30]. Under similar conditions to those we used (non-plasmolyzed yeast cells shaken at 45°C overnight in water with a mass ratio of 0.2 xenobiotic:yeast), an encapsulation efficiency of ~70% has been reported for curcumin [38]. When we were encapsulating the PSIs in yeast, the supernatant during the washing process was colored, meaning that some PSI was not encapsulated, and therefore, was discarded during washing. Therefore, our yeast-encapsulated treatments exposed the larvae to less PSI than we report. Nonetheless, even with less PSI present, yeast-encapsulated PSIs were more toxic to

larvae than non-encapsulated PSIs. Future studies should quantify the yeast encapsulation efficiency to more accurately measure the extent to which yeast encapsulation enhances the larvicidal activity of PSIs.

Many photosensitive molecules are highly effective antimicrobial agents when photoactivated [23–25]. Therefore, a PSI applied to an aquatic environment may be toxic to the microbial community, and in the wild, this damage could harm the general ecosystem. Here, we discovered that yeast encapsulation is a delivery strategy that mitigates the threat to microorganisms while increasing larvicidal efficacy against mosquitoes. We found that yeast-encapsulated curcumin is more larvicidal and offers greater microbial protection when compared to nonencapsulated curcumin. Yeast-encapsulated methylene blue is also more toxic to larvae, but surprisingly, yeast encapsulation does not reduce toxicity against *E. coli* when compared to nonencapsulated methylene blue. This difference could be due to the relative hydrophobicity of these two PSIs. Curcumin, which is so hydrophobic that it is practically insoluble in water at neutral pH [39, 40], may sequester more deeply within the yeast cell whereas methylene blue, which is more hydrophilic [41], may remain closer to the cell surface where ROS can be generated and released into the environment (Fig 5C). Overall, our findings suggest that yeast-encapsulation is a viable strategy to increase the environmental biocompatibility of some PSIs for mosquito control, but not all.

PSIs offer an inexpensive means to control mosquito populations through a mechanism that is unlike other insecticides and less likely to select for resistant mosquito populations [5, 6]. Despite this advantage, the application of non-encapsulated PSIs could have detrimental effects on the surrounding environment, notably to microorganisms. Here we demonstrate that yeast encapsulation can protect microorganisms from PSI toxicity, while also increasing the toxicity against mosquito larvae (Fig 6). Yeast encapsulation of PSIs may have other advantages, such as prolonged storage when desiccated and increased solubility of hydrophobic PSIs, and these should be explored. In summary, yeast encapsulation is a promising delivery strategy that increases both larvicidal efficacy and the biocompatibility of PSIs.

## Supporting information

**S1 Table. Data collected during this study.**
(XLSX)

**S1 Fig. Survival of larvae following exposure to ethanol.** Larvae were incubated in the dark for 2 hr in water and either 0.002% (10 μL), 0.004% (20 μL), 0.01% (50 μL), 0.02% (100 μL), or 0.04% (200 μL) ethanol, for a total volume of 5 mL. Following this incubation, larval survival was measured during 2 hr of photoactivation and once again 22 hr later in ambient lighting (which is insufficient for photoactivation). Whiskers indicate the 95% confidence interval (CI), and n indicates the number of mosquitoes.
(PDF)

**S2 Fig. Survival of larvae following exposure to non-encapsulated or yeast-encapsulated curcumin or methylene blue in the dark.** Larval survival was measured after incubation with either 3 μM curcumin (A), 9 μM curcumin (B), 0.5 μM methylene blue (C), or 1 μM methylene blue (D) that was either non-encapsulated or encapsulated in yeast. Larvae were exposed for 2 hr in continued darkness, followed by an additional 2 hr of darkness and 22 hr of ambient lighting (insufficient for photoactivation). Time zero corresponds the initiation of the second 2 hr darkness incubation. Whiskers indicate the 95% confidence interval (CI), and n indicates the number of mosquitoes.
(PDF)

**S3 Fig. Survival of larvae following exposure to the residual non-encapsulated curcumin and methylene blue remaining in the yeast encapsulate supernatant.** Larval survival was measured following incubation with 50 μL, 200 μL or 500 μL of supernatant from the final wash of the curcumin (A) and methylene blue (B) yeast encapsulate that was added to 5 mL of water. Larvae were exposed for 2 hr in continued darkness, followed by 2 hr of photoactivation and 22 hr of ambient lighting (insufficient for photoactivation). Time zero corresponds to the initiation of the photoperiod. Data were analyzed using the Logrank Mantel Cox Test. Whiskers indicate the 95% confidence interval (CI) and n indicates the number of mosquitoes. (PDF)

**S4 Fig. Survival of *E. coli* following exposure to a 30 min or 2 hr photoperiod.** *E. coli* was either kept in the dark or exposed to a 30 min (A) or 2 hr (B) photoperiod before being plated and grown overnight. Colony forming units (CFUs) were counted to determine bacterial growth and survival. Column heights mark the mean, whiskers denote the S.E.M, and circles are the individual samples. Data for the 30 min Dark treatment is the same data reported as "No PSI" in S5 Fig; the experiments were conducted concurrently. (PDF)

**S5 Fig. Survival of *E. coli* following exposure to yeast-encapsulated curcumin and methylene blue in the dark.** *E. coli* were incubated with no PSI, 9 μM yeast-encapsulated curcumin, or 1 μM yeast-encapsulated methylene blue in the dark for 30 min, before being plated and grown overnight. Colony forming units (CFUs) were counted to determine bacterial growth and survival. Column heights mark the mean, whiskers denote the S.E.M, and squares are the individual samples. Data for the No PSI treatment is the same data reported as 30 min "Dark" in S4A Fig; the experiments were conducted concurrently. (PDF)

## Acknowledgments

The authors thank Jordyn Barr, Lindsay Martin, and Shabbir Ahmed for helpful discussions and feedback on the manuscript.

## Author Contributions

**Conceptualization:** Cole J. Meier, Julián F. Hillyer.

**Data curation:** Cole J. Meier, Veronica R. Wrobleski, Julián F. Hillyer.

**Formal analysis:** Cole J. Meier, Veronica R. Wrobleski, Julián F. Hillyer.

**Funding acquisition:** Julián F. Hillyer.

**Investigation:** Cole J. Meier, Veronica R. Wrobleski.

**Methodology:** Cole J. Meier, Veronica R. Wrobleski, Julián F. Hillyer.

**Project administration:** Cole J. Meier, Julián F. Hillyer.

**Resources:** Cole J. Meier, Julián F. Hillyer.

**Supervision:** Cole J. Meier, Julián F. Hillyer.

**Validation:** Cole J. Meier, Julián F. Hillyer.

**Visualization:** Cole J. Meier, Julián F. Hillyer.

**Writing – original draft:** Cole J. Meier, Julián F. Hillyer.

**Writing – review & editing:** Cole J. Meier, Julián F. Hillyer.

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
