## [Decision Letter · Decision Letter 0]

31 Jul 2024

PONE-D-24-12291Yeast encapsulation of photosensitive insecticides increases toxicity against mosquito larvae while protecting microorganismsPLOS ONE

Dear Dr. Hillyer,

Thank you for submitting your manuscript to PLOS ONE. After careful consideration, we feel that it has merit but does not fully meet PLOS ONE’s publication criteria as it currently stands. Therefore, we invite you to submit a revised version of the manuscript that addresses the points raised during the review process.

**ACADEMIC EDITOR: Please insert comments here and delete this placeholder text when finished.** Be sure to:**Reviewer 1**In the materials and methods, the authors need to clarify few points.Line no. 125 How did the authors discover that pupation doesn't occur? Kindly specify. Did you do any test or Preliminary studies? is this information fr om references?Line no 139 How much is the light or dark time? How many lux ? Kindly add this information.In discussion:Line no 324 Pl modify this phrase. It seems incomplete.Line no. 350 Pl mention in materials and methods regarding microscopic observations**Reviewer 2**Yeast encapsulation of photosensitive insecticides increases toxicity against mosquito larvae while protecting microorganisms

Comments

1. Line 18 - evolution and spread - evolution and subsequent spread

2. Line 23 - However, no effective delivery strategies exist – The statement is quite strong and controversial, change it into currently, there are limited effective delivery strategies...

3. The author can mention few quantitative results, in the abstract.

4. Line 57 - as seen by curcumin not being toxic against adult Danio rerio (zebrafish) – Why sudden comparison with zebrafish?

5. Justify the writing

6. Line 88 - in 4th instar – with 4th instar

7. Line 124 - because this is not a time when larvae pupate, and we discovered that pupation protects larvae from PSI toxicity – change it into a more standard form - as larvae do not pupate during this time, and we discovered that pupation protects larvae from PSI toxicity.

8. Line 100 - either 20 μM methylene blue or 100 μM curcumin were mixed in a 1000 mL Erlenmeyer flask – Why specifically was those 2 concentrations were chosen?

9. Why ethanol is used as a control? What is its significance in this work?

10. Line 200 - we first sought to determine - Therefore, we first aimed to (Keep the language in a standard research article writing tone)

11. Line 291 - does not meaningfully affect - because it does not significantly

12. Line 198 - The phototoxicity of curcumin, however, has only been investigated in Aedes aegypti larvae [7, 16, 17]. – Why only specifically to Aedes aegypti?

13. Line 269 - we suspected that larvae consume more PSIs when they are encapsulated by yeast – Assumptions are not dependable, use proper references, or any occurrence to support this statement.

14. Line 350 - It is possible that the binding of PSIs to yeast components makes them too large to diffuse across the gut epithelium – Cite this

15. Introduction lacks proper transition, from new mosquito strategy, it jumps directly to PSIs, give it a smooth flow.

16. Was there a proper control for the amount of free curcumin or methylene blue that might be released from yeast cells?

17. There’s an assumption that larvae ingest more PSI when encapsulated in yeast. Is there direct evidence like an image of gut content to support this? If yes include it.

18. Different larval stages might respond differently to PSIs. Is there any reason stage 4 were specifically selected?==============================

We look forward to receiving your revised manuscript.

Kind regards,

Muthugounder Subramanian Shivakumar, Ph.D.

Academic Editor

PLOS ONE

Additional Editor Comments:

Dear Author

The reviewers have completed their reviews. they are suggesting a few minor corrections which needs to be carried out in the manuscript. The details of which can be found below:

Reviewer 1

In the materials and methods, the authors need to clarify few points.

Line no. 125 How did the authors discover that pupation doesn't occur? Kindly specify. Did you do any test or Preliminary studies? is this information fr om references?

Line no 139 How much is the light or dark time? How many lux ? Kindly add this information.

In discussion:

Line no 324 Pl modify this phrase. It seems incomplete.

Line no. 350 Pl mention in materials and methods regarding microscopic observations

Reviewer 2

Yeast encapsulation of photosensitive insecticides increases toxicity against mosquito larvae while protecting microorganisms

Comments

1. Line 18 - evolution and spread - evolution and subsequent spread

2. Line 23 - However, no effective delivery strategies exist – The statement is quite strong and controversial, change it into currently, there are limited effective delivery strategies...

3. The author can mention few quantitative results, in the abstract.

4. Line 57 - as seen by curcumin not being toxic against adult Danio rerio (zebrafish) – Why sudden comparison with zebrafish?

5. Justify the writing

6. Line 88 - in 4th instar – with 4th instar

7. Line 124 - because this is not a time when larvae pupate, and we discovered that pupation protects larvae from PSI toxicity – change it into a more standard form - as larvae do not pupate during this time, and we discovered that pupation protects larvae from PSI toxicity.

8. Line 100 - either 20 μM methylene blue or 100 μM curcumin were mixed in a 1000 mL Erlenmeyer flask – Why specifically was those 2 concentrations were chosen?

9. Why ethanol is used as a control? What is its significance in this work?

10. Line 200 - we first sought to determine - Therefore, we first aimed to (Keep the language in a standard research article writing tone)

11. Line 291 - does not meaningfully affect - because it does not significantly

12. Line 198 - The phototoxicity of curcumin, however, has only been investigated in Aedes aegypti larvae [7, 16, 17]. – Why only specifically to Aedes aegypti?

13. Line 269 - we suspected that larvae consume more PSIs when they are encapsulated by yeast – Assumptions are not dependable, use proper references, or any occurrence to support this statement.

14. Line 350 - It is possible that the binding of PSIs to yeast components makes them too large to diffuse across the gut epithelium – Cite this

15. Introduction lacks proper transition, from new mosquito strategy, it jumps directly to PSIs, give it a smooth flow.

16. Was there a proper control for the amount of free curcumin or methylene blue that might be released from yeast cells?

17. There’s an assumption that larvae ingest more PSI when encapsulated in yeast. Is there direct evidence like an image of gut content to support this? If yes include it.

18. Different larval stages might respond differently to PSIs. Is there any reason stage 4 were specifically selected?

Reviewers' comments:

Reviewer's Responses to Questions

**Comments to the Author**

1. Is the manuscript technically sound, and do the data support the conclusions?

Reviewer #1: Yes

Reviewer #2: Yes

2. Has the statistical analysis been performed appropriately and rigorously? 

Reviewer #1: Yes

Reviewer #2: Yes

3. Have the authors made all data underlying the findings in their manuscript fully available?

Reviewer #1: No

Reviewer #2: Yes

4. Is the manuscript presented in an intelligible fashion and written in standard English?

Reviewer #1: Yes

Reviewer #2: Yes

5. Review Comments to the Author

Reviewer #1: In the materials and methods, the authors need to clarify few points.

Line no. 125 How did the authors discover that pupation doesn't occur? Kindly specify. Did you do any test or Preliminary studies? is this information fr om references?

Line no 139 How much is the light or dark time? How many lux ? Kindly add this information.

In discussion:

Line no 324 Pl modify this phrase. It seems incomplete.

Line no. 350 Pl mention in materials and methods regarding microscopic observations.

Reviewer #2: Yeast encapsulation of photosensitive insecticides increases toxicity against mosquito larvae while protecting microorganisms

Comments

1. Line 18 - evolution and spread - evolution and subsequent spread

2. Line 23 - However, no effective delivery strategies exist – The statement is quite strong and controversial, change it into currently, there are limited effective delivery strategies...

3. The author can mention few quantitative results, in the abstract.

4. Line 57 - as seen by curcumin not being toxic against adult Danio rerio (zebrafish) – Why sudden comparison with zebrafish?

5. Justify the writing

6. Line 88 - in 4th instar – with 4th instar

7. Line 124 - because this is not a time when larvae pupate, and we discovered that pupation protects larvae from PSI toxicity – change it into a more standard form - as larvae do not pupate during this time, and we discovered that pupation protects larvae from PSI toxicity.

8. Line 100 - either 20 μM methylene blue or 100 μM curcumin were mixed in a 1000 mL Erlenmeyer flask – Why specifically was those 2 concentrations were chosen?

9. Why ethanol is used as a control? What is its significance in this work?

10. Line 200 - we first sought to determine - Therefore, we first aimed to (Keep the language in a standard research article writing tone)

11. Line 291 - does not meaningfully affect - because it does not significantly

12. Line 198 - The phototoxicity of curcumin, however, has only been investigated in Aedes aegypti larvae [7, 16, 17]. – Why only specifically to Aedes aegypti?

13. Line 269 - we suspected that larvae consume more PSIs when they are encapsulated by yeast – Assumptions are not dependable, use proper references, or any occurrence to support this statement.

14. Line 350 - It is possible that the binding of PSIs to yeast components makes them too large to diffuse across the gut epithelium – Cite this

15. Introduction lacks proper transition, from new mosquito strategy, it jumps directly to PSIs, give it a smooth flow.

16. Was there a proper control for the amount of free curcumin or methylene blue that might be released from yeast cells?

17. There’s an assumption that larvae ingest more PSI when encapsulated in yeast. Is there direct evidence like an image of gut content to support this? If yes include it.

18. Different larval stages might respond differently to PSIs. Is there any reason stage 4 were specifically selected?

6. PLOS authors have the option to publish the peer review history of their article (what does this mean?). If published, this will include your full peer review and any attached files.

Reviewer #1: **Yes: **Yashkamal K.

Reviewer #2: **Yes: **Chinnaperumal Kamaraj

---

## [Author Response · Author response to Decision Letter 0]

5 Aug 2024

• Additional Editor Comments:

Dear Author

The reviewers have completed their reviews. they are suggesting a few minor corrections which needs to be carried out in the manuscript. The details of which can be found below:

Author response: Thank you for serving as the editor of this manuscript. Below we address the reviewer comments and specify how we modified the manuscript. We hope this revision can result in the acceptance of this manuscript in PLOS ONE.

• Reviewer 1

In the materials and methods, the authors need to clarify few points.

Line no. 125 How did the authors discover that pupation doesn't occur? Kindly specify. Did you do any test or Preliminary studies? is this information fr om references?

Author response: This sentence references a paper we published last year in Parasites & Vectors (reference 35): “Mosquito larvae exposed to a sublethal dose of photosensitive insecticides have altered juvenile development but unaffected adult life history traits”. In that manuscript, we show that pupation only occurs in the evening (Additional File 3: Fig. S2) and that pupation protects larvae from PSI toxicity (Table 1). 

Line no 139 How much is the light or dark time? How many lux ? Kindly add this information.

Author response: The times for light and dark are detailed throughout the manuscript, including in the x-axis of many of the figures. We now specify in the “Larval incubation, photoactivation, and survival” section (2nd paragraph) of the methods that illumination is ~1,250 lux.

In discussion:

Line no 324 Pl modify this phrase. It seems incomplete.

Author response: The sentence starting in line 324 reads, “Overall, yeast encapsulation is a promising strategy to increase the larvicidal efficacy and environmental biocompatibility of some PSIs to control mosquito populations”. To us, this reads like a complete sentence.

Line no. 350 Pl mention in materials and methods regarding microscopic observations

Author response: We added the following two sentences to the second section of the methods: “For microscopic examinations, larvae were viewed under oblique coherent contrast trans-illumination using a Nikon SMZ 1500 stereomicroscope (Nikon Corporation, Tokyo, Japan). Micrographs were acquired using a Nikon Digital Sight DS-Fi1 5 MP CCD Color Camera and Nikon's Advanced Research NIS Elements software.”

• Reviewer 2

1. Line 18 - evolution and spread - evolution and subsequent spread

Author response: Modified as suggested.

2. Line 23 - However, no effective delivery strategies exist – The statement is quite strong and controversial, change it into currently, there are limited effective delivery strategies...

Author response: We deleted the sentence.

3. The author can mention few quantitative results, in the abstract.

Author response: We believe that this is a matter of style. We prefer to use the abstract to convey the message, and let the reader visit the entire manuscript for the quantitative results.

4. Line 57 - as seen by curcumin not being toxic against adult Danio rerio (zebrafish) – Why sudden comparison with zebrafish?

Author response: We reference another a paper by a different research group that demonstrates curcumin is not toxic to zebrafish. The comparison is because of what the other investigators did; we did not choose the organism. 

5. Justify the writing

Author response: We do not understand what is being asked in this comment.

6. Line 88 - in 4th instar – with 4th instar

Author response: Modified as suggested.

7. Line 124 - because this is not a time when larvae pupate, and we discovered that pupation protects larvae from PSI toxicity – change it into a more standard form - as larvae do not pupate during this time, and we discovered that pupation protects larvae from PSI toxicity.

Author response: As suggested, the sentence was modified to “…because pupation is minimal during this time, and we discovered that pupation protects larvae from PSI toxicity”.

8. Line 100 - either 20 μM methylene blue or 100 μM curcumin were mixed in a 1000 mL Erlenmeyer flask – Why specifically was those 2 concentrations were chosen?

Author response: The concentrations were selected based on preliminary experiments. But for the purpose of the experiments this is unimportant. What is important is that the YEPSI solutions were clear after washing and that we list the concentrations that were experimentally used.

9. Why ethanol is used as a control? What is its significance in this work?

Author response: The significance of ethanol is explained in the first section of the results: “Due to curcumin’s poor solubility in water [36, 37], we prepared curcumin stock solutions in ethanol. Therefore, before measuring the phototoxicity of non-encapsulated curcumin, we set to determine the maximum amount of ethanol that larvae can tolerate.”

10. Line 200 - we first sought to determine - Therefore, we first aimed to (Keep the language in a standard research article writing tone)

Author response: Modified as suggested.

11. Line 291 - does not meaningfully affect - because it does not significantly

Author response: Modified as suggested.

12. Line 198 - The phototoxicity of curcumin, however, has only been investigated in Aedes aegypti larvae [7, 16, 17]. – Why only specifically to Aedes aegypti?

Author response: The reason is that other investigators have only investigated Aedes aegypti. Here we add to current data by investigating Anopheles gambiae.

13. Line 269 - we suspected that larvae consume more PSIs when they are encapsulated by yeast – Assumptions are not dependable, use proper references, or any occurrence to support this statement.

Author response: Key to science is intellectual curiosity that leads to hypotheses and predictions. If something is known, then most likely it is not worth doing again. Here, we had a prediction, and we tested it. The outcome is presented in figure 4. We do not see a problem with our statement.

14. Line 350 - It is possible that the binding of PSIs to yeast components makes them too large to diffuse across the gut epithelium – Cite this

Author response: We write, “It is possible that the binding of PSIs to yeast components makes them too large to diffuse across the gut epithelium”. Here we are proposing a possible mechanism (“it is possible…”). We do not understand the criticism.

15. Introduction lacks proper transition, from new mosquito strategy, it jumps directly to PSIs, give it a smooth flow.

Author response: The first paragraph of the introduction ends “new mosquito control strategies are desperately needed”. The second paragraph proposes a strategy: “One promising option for mosquito control is the use of photosensitive insecticides”. We hold that the flow is smooth as is.

16. Was there a proper control for the amount of free curcumin or methylene blue that might be released from yeast cells?

Author response: We control for this in S3 Fig: “Survival of larvae following exposure to the residual non-encapsulated curcumin and methylene blue remaining in the yeast encapsulate supernatant.”

17. There’s an assumption that larvae ingest more PSI when encapsulated in yeast. Is there direct evidence like an image of gut content to support this? If yes include it.

Author response: Figure 4 presents qualitative evidence.

18. Different larval stages might respond differently to PSIs. Is there any reason stage 4 were specifically selected?

Author response: We added the following sentence to the methods section: “Fourth instar larvae were selected for experimentation because of their larger size, and so that the findings could be correlated to our previously published research”.

---

## [Decision Letter · Decision Letter 1]

27 Aug 2024

Yeast encapsulation of photosensitive insecticides increases toxicity against mosquito larvae while protecting microorganisms

PONE-D-24-12291R1

Dear Dr. Hillyer,

We’re pleased to inform you that your manuscript has been judged scientifically suitable for publication and will be formally accepted for publication once it meets all outstanding technical requirements.

Kind regards,

Muthugounder Subramanian Shivakumar, Ph.D.

Academic Editor

PLOS ONE

Additional Editor Comments (optional):

the manuscript can be accepted

Reviewers' comments:

Reviewer's Responses to Questions

**Comments to the Author**

1. If the authors have adequately addressed your comments raised in a previous round of review and you feel that this manuscript is now acceptable for publication, you may indicate that here to bypass the “Comments to the Author” section, enter your conflict of interest statement in the “Confidential to Editor” section, and submit your "Accept" recommendation.

Reviewer #1: (No Response)

2. Is the manuscript technically sound, and do the data support the conclusions?

Reviewer #1: Yes

3. Has the statistical analysis been performed appropriately and rigorously? 

Reviewer #1: Yes

4. Have the authors made all data underlying the findings in their manuscript fully available?

Reviewer #1: Yes

5. Is the manuscript presented in an intelligible fashion and written in standard English?

Reviewer #1: Yes

6. Review Comments to the Author

Reviewer #1: The work is interesting and in my opinion the authors have adequately addressed the queries raised during the previous review.

7. PLOS authors have the option to publish the peer review history of their article (what does this mean?). If published, this will include your full peer review and any attached files.

Reviewer #1: **Yes: **Yashkamal K

---

## [Editor Report · Acceptance letter]

30 Aug 2024

PONE-D-24-12291R1 

PLOS ONE

Dear Dr. Hillyer, 

I'm pleased to inform you that your manuscript has been deemed suitable for publication in PLOS ONE. Congratulations! Your manuscript is now being handed over to our production team.

Kind regards, 

on behalf of

Dr. Muthugounder Subramanian Shivakumar 

Academic Editor

PLOS ONE